# Posterior Attention Models for Sequence to Sequence Learning

**Shiv Shankar**
University of Massachusetts Amherst
sshankar@umass.edu

**Sunita Sarawagi**
IIT Bombay
sunita@iitb.ac.in

## Abstract

Modern neural architectures critically rely on attention for mapping structured inputs to sequences. In this paper we show that prevalent attention architectures do not adequately model the dependence among the attention and output tokens across a predicted sequence. We present an alternative architecture called Posterior Attention Models that after a principled factorization of the full joint distribution of the attention and output variables, proposes two major changes. First, the position where attention is marginalized is changed from the input to the output. Second, the attention propagated to the next decoding stage is a posterior attention distribution conditioned on the output. Empirically on five translation and two morphological inflection tasks the proposed posterior attention models yield better BLEU score and alignment accuracy than existing attention models.

## 1 Introduction

Attention is a critical module of modern neural models for sequence to sequence learning as applied to tasks like translation, grammar error correction, morphological inflection, and speech to text conversion. Attention specifies what part of the input is relevant for each output. Many variants of attention have been proposed including soft (Bahdanau et al., 2014; Luong et al., 2015), sparse (Martins & Astudillo, 2016), local (Luong et al., 2015), hard (Xu et al., 2015; Zaremba & Sutskever, 2015), monotonic hard (Yu et al., 2016; Aharoni & Goldberg, 2017), hard non-monotonic (Wu et al., 2018; Shankar et al., 2018), and variational (Deng et al., 2018), The most prevalent of these is soft attention that computes attention for each output as a multinomial distribution over the input states. The multinomial probabilities serve as weights, and an attention weighted sum of input states serves as relevant context for the output and subsequent attention. Soft attention is end to end differentiable, easy to implement, and hence widely popular. Hard attention and sparse attentions are difficult to implement and not popularly used.

In this paper we revisit the statistical soundness of the way soft attention and other variants capture the dependence between attention and output variables, and among multiple attention variables along the length of the sequence. Our investigation leads to a more principled model that we call the Posterior Attention Model (PAM). We start with an explicit joint distribution of all output and attention variables in a predicted sequence. We then propose a tractable approximation that retains the advantages of forward dependence and token-level decomposition and thus leads to efficient training and inference. The computations performed at each decode step has two important differences with existing models. First, at each decoding step the probability of the output token is a mixture of output probability for each attention. In contrast, existing models take a mixture of the input, and compute a single output distribution from this diffused mixed input. We show that our direct coupling of output and attention gives the benefit of hard attention without its computational challenges. Second, we introduce the notion of a *posterior* attention distribution, that is, the attention distribution conditioned on the current output. We show that it is both statistically sounder and more accurate to condition subsequent attention on the output corrected posterior attention, rather than the output independent prior attention as in existing models.

We evaluate the posterior attention model on five translation tasks and two morphological inflection tasks. We show that posterior attention provides improved BLEU score, higher alignment accuracy, and better input coverage. We also empirically analyze the reasons behind the improved performance

of the posterior attention model. We discover that the entropy of posterior attention is much lower than entropy of soft attention. This is a significant finding that challenges the current practice of computing attention distribution without considering the output token. The running time overhead of posterior attention is only 40% over existing soft-attention.

## 2 JOINT DISTRIBUTION FOR ATTENTION AND OUTPUT VARIABLES

Our goal is to model the conditional distribution $\Pr(\mathbf{y}|\mathbf{x})$ of an output sequence $\mathbf{y} = y_1, \ldots, y_n$ given an input sequence $\mathbf{x} = x_1, \ldots, x_m$. Each output $y_t$ is a discrete token from a typically large vocabulary $V$. Each $x_j$ can be any abstract input. Typically a RNN encodes the input sequence into a sequence of state vectors $\mathbf{x}_1, \ldots, \mathbf{x}_m$, which we jointly denote as $\mathbf{x}_{1:m}$. Each $y_t$ depends not only on other tokens in the sequence, but on some specific focused part of the input sequence. A hidden variable $a_t$, called the attention variable, denotes which part of $\mathbf{x}_{1:m}$ the output $y_t$ depends on. We denote the set of all attention as $\mathbf{a} = a_1, \ldots, a_n$. During training the input $\mathbf{x}$ and output $\mathbf{y}$ are observed but the attention $\mathbf{a}$ is hidden. Hence, we write $\Pr(\mathbf{y}|\mathbf{x})$ as

$$\Pr(\mathbf{y}|\mathbf{x}_{1:m}) = \sum_{\mathbf{a}} \Pr(\mathbf{y}, \mathbf{a}|\mathbf{x}_{1:m}) = \sum_{a_1, \ldots, a_n} \Pr(y_1, \ldots, y_n, a_1, \ldots, a_n|\mathbf{x}_{1:m}) \tag{1}$$

The number of variables involved in this summation is daunting, and we need to approximate. We first review how existing soft attention-based encoder decoder models handle this challenge.

### 2.1 EXISTING ATTENTION-BASED ENCODER DECODER MODEL

Existing Encoder-Decoder (ED) networks factorize $\Pr(\mathbf{y}|\mathbf{x}_{1:m})$ by applying chain rule on $\mathbf{y}$ variables as $\prod_{t=1}^{n} \Pr(y_t|\mathbf{x}_{1:m}, y_1, \ldots, y_{t-1})$. A decoder RNN summarizes the variable length history $y_1, \ldots, y_{t-1}$ as a decoder state $\mathbf{s}_t$, so that $\Pr(\mathbf{y}|\mathbf{x}_{1:m}) = \prod_{t=1}^{n} \Pr(y_t|\mathbf{x}_{1:m}, \mathbf{s}_t)$. The distribution of each attention variable $a_t$ is computed as a function of the decoder state and encoder state as: $\Pr(a|\mathbf{x}_{1:m}, \mathbf{s}_t) \propto e^{A_\theta(\mathbf{x}_a, \mathbf{s}_t)}$. Here $A_\theta(.,.)$ is an end-to-end trained function of input state $\mathbf{x}_a$ and decoder state $\mathbf{s}_t$. We will use the short form $P_t(a)$ for $\Pr(a_t|\mathbf{x}_{1:m}, \mathbf{s}_t)$. Thereafter, an attention weighted sum of the input states $\sum_a P_t(a)\mathbf{x}_a$ called input context $\mathbf{c}_t$ is computed. The distribution of $y_t$ is computed from $\mathbf{c}_t$ (capturing attention) and $\mathbf{s}_t$ capturing previous $y$ as:

$$\Pr(\mathbf{y}|\mathbf{x}_{1:m}) = \prod_{t=1}^{n} \Pr(y_t|\mathbf{s}_t, \sum_{a_t} P_t(a_t)\mathbf{x}_{a_t}) \tag{2}$$

Next, $\mathbf{c}_t$ is fed to the decoder RNN along with $y_t$ for computing the next state: $\mathbf{s}_{t+1} = \text{RNN}(\mathbf{s}_t, \mathbf{c}_t, y_t)$. Figure 1[left] summarizes the compute equations of the encoder-decoder model. If we view Equation 2 as an approximation of the full joint distribution in Equation 1, we find that the treatment of the attention variables has been rather ad hoc. Attention was introduced as an after-thought of factorizing on the $y_t$ variables, the interaction among various $a_t$s is not expressed, and the influence of $a_t$ on $y_t$ by diffusing the inputs is unprincipled.

### 2.2 LATENT ATTENTION MODELS

There have been a number of recent works (Wu et al., 2018; Deng et al., 2018; Shankar et al., 2018) which model attention as latent-alignment variable in the joint distribution Equation 1. The model becomes more tractable by assuming that the output $y_t$ at each step is dependent only on $a_t$ and previous outputs $\mathbf{y}_{<t}$ i.e. $P(y_t|\mathbf{y}_{<t}, \mathbf{a}_{<t}, a_t) = P(y_t|\mathbf{y}_{<t}, a_t)$. Both Shankar et al. (2018); Wu et al. (2018) further assume that attention at each time step is independent of attention at other timesteps, and marginalize over all attentions at each time-step as in Equation 3. Deng et al. (2018) also rely on the same assumption but instead of direct marginalization use variational methods. All these models can be considered as a neuralization of IBM Model 1.

$$\Pr(\mathbf{y}|\mathbf{x}_{1:m}) = \prod_{t=1}^{n} \sum_{a_t} P_t(a_t) \Pr(y_t|\mathbf{s}_t, \mathbf{x}_{a_t}) \tag{3}$$

Such mean-field assumption while making the model significantly simpler ignore relationships between attentions which is undesirable. Moreover as we will see in the experiments, they also ignore consistency between attention and output variables. We next present a principled model of the interaction of the various attention and output variables, which is as efficient as the mean field factorization approach while allowing more realistic latent behavior. We call our proposed approach: Posterior Attention Models or PAM.

## 2.3 Posterior Attention Models

Our goal is to express the joint distribution as a product of tractable terms computed at each time step much like in existing ED model, but via a less ad hoc treatment of the attention variables $a_1, \ldots, a_n$. We use $\mathbf{y}_{<t}, \mathbf{a}_{<t}$ to denote all output and attention variables before $t$ that is, $y_1, , \ldots y_{t-1}, a_1, \ldots a_{t-1}$. Here and in the rest of the paper we will drop $\mathbf{x}_{1:m}$ to use the shorter form $P(\mathbf{y})$ for $\Pr(\mathbf{y}|\mathbf{x}_{1:m})$. We first factorize Eq 1 via chain rule, like in ED but jointly on both $\mathbf{a}$ and $\mathbf{y}$.

$$P(\mathbf{y}) = \sum_{\mathbf{a}} P(\mathbf{y}, \mathbf{a}) = \sum_{\mathbf{a}_{<n}, a_n} P(y_n|\mathbf{y}_{<n}, \mathbf{a}_{<n}, a_n) P(a_n|\mathbf{y}_{<n}, \mathbf{a}_{<n}) P(\mathbf{a}_{<n}|\mathbf{y}_{<n}) P(\mathbf{y}_{<n})$$

We then make two mild assumptions: First, the same local attention assumption that $y_t$ is dependent only on $a_t$ and previous outputs $\mathbf{y}_{<t}$ as detailed earlier. Second, a Markovian assumption on the attention variables i.e., $P(a_n|\mathbf{y}_{<n}, \mathbf{a}_{<n}) = P(a_n|\mathbf{y}_{<n}, a_{n-1})$.

These together allows us to simplify the above joint as:

$$P(\mathbf{y}) = P(\mathbf{y}_{<n}) \sum_{a_n} P(y_n|\mathbf{y}_{<n}, a_n) \sum_{a_{n-1}} P(a_n|\mathbf{y}_{<n}, a_{n-1}) P(a_{n-1}|\mathbf{y}_{<n})$$

$$= \prod_{t=1}^{n} \sum_{a_t} P(y_t|\mathbf{y}_{<t}, a_t) \sum_{a_{t-1}} P(a_t|a_{t-1}, \mathbf{y}_{<t}) P(a_{t-1}|\mathbf{y}_{<t})$$

The last equality is after applying the same rewrite recursively on $P(\mathbf{y}_{<n})$. Thus, we have expressed the joint distribution as a product of factors that apply at each decoding step $t$ while conditioning only on previous outputs and attention. The term $\sum_{a_{t-1}} P(a_t|a_{t-1}, \mathbf{y}_{<t}) P(a_{t-1}|\mathbf{y}_{<t}) = P(a_t|\mathbf{y}_{<t})$ is the attention at step '$t$' conditioned on all previous outputs. For reasons that will soon become clear we call this the **prior attention** at $t$ and denote as $\text{Prior}_t(a)$. We call $P(a_{t-1}|\mathbf{y}_{<t}) = P(a_{t-1}|\mathbf{y}_{<(t-1)}, y_{t-1})$ as the **posterior attention** $\text{Postr}(a_{t-1})$ since this is the attention distribution *after observing the output label* at the corresponding step, unlike in prior attention. We expect this attention to be more accurate than the prior that is computed without knowledge of the output token at that step. We compute posterior attention at any $t$ using prior attention at $t-1$ by applying Bayes rule as follows:

$$\text{Postr}_t(a_t) = P(a_t|\mathbf{y}_{<t}, y_t) = \frac{P(y_t|\mathbf{y}_{<t}, a_t) P(a_t|\mathbf{y}_{<t})}{P(y_t|\mathbf{y}_{<t})} = \frac{P(y_t|\mathbf{y}_{<t}, a_t) \text{Prior}_t(a_t)}{P(y_t|\mathbf{y}_{<t})} \quad (4)$$

$$\text{Prior}_t(a_t) = \sum_{a_{t-1}} P(a_t|\mathbf{y}_{<t}, a_{t-1}, y_{t-1}) \text{Postr}_{t-1}(a_{t-1}) \quad (5)$$

The above equations give us the important result that the attention at step $t$ should be computed from the posterior attention of the previous step. Intuitively, also it makes sense because attention reflects an alignment of the input and output, and its distribution will improve if the output is known. We get into details of computing such coupled attention in Section 2.3.1.

### 2.3.1 Computation of Prior Attention Distribution

We use the RNN to summarize $\mathbf{y}_{<t}$ as a fixed length vector $\mathbf{s}_t$ as in current ED models. We then discuss three different methods we explored for computing $\sum_{a_{t-1}} P(a_t|\mathbf{s}_t, a_{t-1}, y_{t-1}) \text{Postr}_{t-1}(a_{t-1})$. Our methods are designed to be light-weight in terms of the number of extra parameters they consume beyond the default soft-attention methods to have the fairest comparison.

**Postr-Joint** The simplest of these uses the same decoder RNN to absorb the posterior attention of the previous step. We linearize the function using the first order Taylor expansion to efficiently approximate computation of $\text{Prior}_t(a)$ similar to the deterministic technique of Xu et al. (2015)

$$\text{Prior}_t(a_t) = \sum_{a'} P(a_t|\mathbf{s}_{t-1}, y_{t-1}, a') \text{Postr}_{t-1}(a') \approx P(a_t|\mathbf{s}_{t-1}, y_{t-1}, \sum_{a'} \text{Postr}_{t-1}(a') x_{a'}) \quad (6)$$

The above equation suggests that the decoder RNN state should be updated as $\mathbf{s}_t = \text{RNN}(\mathbf{s}_{t-1}, \sum_{a'} \text{Postr}_{t-1}(a') x_{a'}, y_{t-1})$. The computation here is thus similar to existing ED model's but the crucial difference is that the context used to update the RNN is computed from posterior attention, and not the prior attention. We will see that this leads to large improvement in accuracy.

**Proximity biased** Next we experiment with models that explicitly couple adjacent attention. These models utilize an index based coupling between attention positions of the form

$$\text{Prior}_t(a_t) = \sum_{a'} P(a_t|\mathbf{s}_{t-1}, y_{t-1}, a')\text{Postr}_{t-1}(a') \approx \sum_{a'} P(a_t|\mathbf{s}_t, a')\text{Postr}_{t-1}(a')$$

$$= \sum_{a'} \text{Postr}_{t-1}(a')\frac{\exp\left(k(a_t, a'_{t-1}) + A_\theta(x_{a_t}, \mathbf{s}_t)\right)}{Z_{a'}}$$

where $A_\theta(x_{a_t}, \mathbf{s}_t)$ is the attention logit computed from the previous RNN step, $k(a_t, a_{t-1})$ is the attention coupling energy and $Z$ is the normalization constant.

In the proximity based attention the coupling energy $k(a_t, a_{t-1})$ is given by $\mathbb{I}(|a_t - a_{t-1}| < 3)\delta_{a_t - a_{t-1}}$. This model provides a greater focus on attending states within a window of size five centered around the recently attended input state. We label this model as Prox-Postr-Joint in our experiments.

**Monotonicity biased**     This method differs from the above proximity-based attention only in how it defines the coupling energy $k(a_t, a_{t-1})$. As the name implies, in this method $k(a_t, a_{t-1})$ is a monotonic energy given by $\mathbb{I}(a_t > a_{t-1})\delta^{a_t - a_{t-1} - 1}$. This model provides a positive exponentially decaying bias towards encoder states which are to the right of the current attended state, thus influecning attention to be more monotonic. As we shall see tasks with natural monotonic attention benefit from this form of bias. This model is denoted as Mono-Postr-Joint in our experiments.

## 2.4   PUTTING IT ALL TOGETHER

In Figure 1 we put together the final set of equations that are used to compute the output distribution and contrast with existing attention model. We call this overall architecture as Posterior Attention Model (PAM). First note that in PAM, we explicitly compute a joint distribution of output and attention at each step and marginalize out the attention. Thus, the output is a mixture of multiple output distributions each of which is a function of one focused input (like in hard attention), and not a diffused sum of the input (like in soft attention). This difference in the way attention is marginalized is not only statistically sound, but also leads to higher accuracy. The only downside of the joint model is that we need to compute $m$ softmaxes for each output $y_t$, and this may be impractical when the vocabulary size is large. A simple and effective fix to this is to select the top-K attentions based on $\text{Prior}_t$ and compute the final output distribution as.

$$\sum_a P(y_t|\mathbf{s}_t, x_a)\text{Prior}_t(a) \approx \sum_{a \in \text{TopK}(\text{Prior}_t(a))} \text{Prior}_t(a)P(y_t|\mathbf{s}_t, \mathbf{x}_a) \tag{7}$$

Small values of $K$ (order 6), suffice to provide good performance The second difference is that the attention distribution that is propagated to the next step is posterior to observing the current output. We derived this from a principled rewrite of the joint distribution, and were pleasantly surprised to see significant accuracy gains by this subtle difference in the way the decoder state is updated. Computing the posterior attention does not incur any additional overheads because the joint attention-output distribution was already materialized in the first equation. However, due to the sparsity induced by the top-K operation on attention probabilities, the posterior probabilities are unrealistically sparse. As such we augment the posterior attention using input from standard attention, by using an equally weighted combination of the two distributions. Third, the prior attention distribution is explicitly conditioned on the previous attention. This allowed us to incorporate various application-specific natural biases like proximity and monotonicity of adjacent attentions.

## 2.5   ALTERNATIVE REWRITES

Our rewrite although somewhat unconventional was derived to satisfy two important goals: First, explain the need to propagate posterior attention to subsequent steps. Second, to factorize the joint as the *product* of the local distribution at each time $t$ which allows efficient gradient updates and minimal changes to existing beam-search inference. A more conventional rewrite for handling Markovian dependencies is the standard forward algorithm which works as follows. First we write:

$$p(\mathbf{y}) = \sum_{\mathbf{a}} \prod_{t=1}^{n} p(y_t|\mathbf{y}_{<t}, a_t)p(a_t|\mathbf{y}_{<t}, a_{t-1}).$$

$$\Pr(\mathbf{y}|\mathbf{x}_{1:m}) = \prod_{t=1}^{n} \Pr(y_t|\mathbf{s}_t, \sum_{a=1}^{m} P_t(a)\mathbf{x}_a) \quad (8)$$

$$\mathbf{s}_{t+1} = \text{RNN}(\mathbf{s}_t, y_t, \sum_a P_t(a)\mathbf{x}_a) \quad (9)$$

$$P_t(a) = \frac{e^{A_\theta(\mathbf{x}_a, \mathbf{s}_t)}}{\sum_{r=1}^{m} e^{A_\theta(\mathbf{x}_r, \mathbf{s}_t)}} \quad (10)$$

$$\Pr(\mathbf{y}|\mathbf{x}_{1:m}) = \prod_{t=1}^{n} \sum_{a=1}^{m} P(y_t|\mathbf{s}_t, \mathbf{x}_a)\text{Prior}_t(a) \quad (11)$$

$$\mathbf{s}_{t+1} = \text{RNN}(\mathbf{s}_t, y_t, \sum_a \text{Postr}_t(a)\mathbf{x}_a) \quad (12)$$

$$\text{Postr}_t(a) = \frac{P(y_t|\mathbf{s}_t, x_a)\text{Prior}_t(a)}{\sum_{a'} P(y_t|\mathbf{s}_t, x_{a'})\text{Prior}_t(a')} \quad (13)$$

$$\text{Prior}_t(a_t) = \sum_{a'} P(a_t|\mathbf{s}_{t-1}, a')\text{Postr}_{t-1}(a') \quad (14)$$

$$P(a_t|\mathbf{s}_{t-1}, a') = \text{See Section 2.3.1} \quad (15)$$

Figure 1: Comparing the Equations for computing $\Pr(\mathbf{y}|\mathbf{x}_{1:m})$ of existing encoder decoder model based on soft attention (Left) with our Posterior Attention Model (Right)

Then use the forward algorithm to compute:

$$\alpha_t(a) = P(a_t = a, \mathbf{y}_{<=t}) = p(y_t|\mathbf{s}_t, a_t = a)\sum_{a'} \alpha_{t-1}(a')p(a_t = a|s_t, a_{t-1} = a').$$

which then gives the joint distribution as $P(\mathbf{y}) = \sum_a \alpha_n(a)$. This expression is neither in the outer product form, nor does it motivate the need for posterior attention.

## 3  RELATED WORK

The de facto standard for sequence to sequence learning via neural networks is the encoder decoder model. Ever since their first introduction in Bahdanau et al. (2014), many different attention models have been proposed. We discuss them here.

**Soft Attention** is the de-facto mechanism for seq2seq learning et al (2018). It was proposed for translation in Bahdanau et al. (2014) and refined further in Luong et al. (2015). The output derives from an attention averaged context. The advantage is end to end differentiability.

**Hard Attention** was proposed in Xu et al. (2015) and attends to exactly one input state for an output. The merit of hard attention is that the output is determined from a single input rather than an average of all inputs. Accordingly it has proven useful in when explicit focus is beneficial such as model adaptation Shankar & Sarawagi (2018) and catastrophic forgetting Serrà et al. (2018). However due to non-differentiability, training Hard-Attention requires the REINFORCE Williams (1992) algorithm and is subject to high variance, requiring careful tricks to train reliably. Yu et al. (2016) keep the encoder and decoder independent to allow for easier marginalization. Aharoni & Goldberg (2017) use a monotonic hard attention and avoid the problem, by supervising hard attention with external alignment information. Our model in equation 11 uses hard attention on the encoder states. However unlike standard hard attention we do not use a one-hot attention and instead are computing the exact marginalization.

**Sparse/Local Attention** Many attempts have been made to bridge the gap between soft and hard attention. Luong et al. (2015) proposes local attention that averages a window of input. This has been refined later to include syntax (Chen et al., 2017; Sennrich & Haddow, 2016; Chen et al., 2018) and has been explored for image captioning in Gregor et al. (2015). A related idea to hard attention is to make it sparse using sparsity inducing operators (Martins & Astudillo, 2016; Niculae & Blondel, 2017). However, all sparse/local attention methods continue to compute $P(y)$ from an attention weighted sum of inputs like in soft attention.

**Recurrent Attention** Yang et al. (2016) have previously modeled relationship between the attentions at different time steps by using a recurrent history mechanism. The attention history of an input word and its surrounding words are captured in a summary vector by an RNN, which is provided as further input to the attention mechanism for incorporating dependence on history. While both works model dependence between attention at different steps, our principled rewrite of the joint distribution shows that posterior attention should be the link to the next attention.

**Latent Attention Models** Our model can be considered as a generalization of the work of Wang et al. (2018) to the case where attention is also provided to the RNN. The model in Shankar et al. (2018)

also factorizes the joint distribution and are identical to our Prior-Joint model. However unlike these models we explicitly model the posterior attention distribution and attention coupling. Deng et al. (2018) proposes to learn the posterior attention via variational methods. A key difference is while their model tries to *supervise attention* using a posterior inference network via a KL term, we directly *use the actual posterior* for computing attention in the subsequent steps. In our method, posterior attention is used in identical roles across training and inference, unlike in Deng et al. (2018)'s that rely on variational training.

**Structured Attention Networks** Similar to this work, Kim et al. (2017) interpret attention as latent structural variable. The authors then take advantage of easy inference in certain graphical models to implement forms of segmental and syntactic attention. These works only focus on attention at each step independently whereas our focus is modeling the dependency among adjacent attention. Moreover our posterior attention framework is independent of how the prior attention at each position is modeled. In this paper we assumed a multinomial distribution but the structured distribution of Kim et al. (2017) can also benefit from our posterior coupling.

## 4 EXPERIMENTS

We compare our posterior attention model on two sequence to sequence learning tasks: machine translation and morphological inflection. We compare on the following methods:

**Soft:** attention Luong et al. (2015).

**Sparse:** attention Niculae & Blondel (2017).

**Postr-Joint:** Our default PAM ( 2.3.1).

**Mono-Postr-Joint:** monotonic PAM (§ 2.3.1)

**Prox-Postr-Joint:** Proximity biased PAM(§ 2.3.1)

**Prior-Joint:** suppresses the dependence on previous attention like in current ED models, thus reducing to methods proposed in Shankar et al. (2018); Wu et al. (2018) .

**Variational:** method of Deng et al. (2018) using author's code with our tokenization and architecture.

### 4.1 MACHINE TRANSLATION

We experiment on five language pairs from three datasets: **IWSLT15 English↔Vietnamese**, **IWSLT14 German↔English** Cettolo et al. (2015); and **WAT17 Japanese→English** Nakazawa et al. (2016). We use a 2 layer bi-directional encoder and 2 layer decoder with 512 LSTM units and 0.2 dropout with vanilla SGD optimizer. We use word level encoding for all translation tasks.

| Dataset | Attention | PPL | BLEU | |
|---|---|---|---|---|
| | | | B=4 | B=10 |
| IWSLT14 DE-EN | Soft | 9.61 | 28.6 | 28.5 |
| | Sparse | 9.85 | 28.4 | 28.0 |
| | Variational | | 29.8 | 29.8 |
| | Prior-Joint | 8.47 | 29.7 | 29.6 |
| | Postr-Joint | 8.51 | 29.8 | 29.7 |
| | Mono-Postr-Joint | **8.23** | **30.0** | 29.9 |
| | Prox-Postr-Joint | 8.26 | 29.8 | 29.7 |
| IWSLT14 EN-DE | Soft | 10.68 | 24.2 | 24.2 |
| | Sparse | 10.89 | 23.4 | 23.3 |
| | Variational | | 25.2 | 25.3 |
| | Prior-Joint | 8.72 | 25.4 | 25.3 |
| | Postr-Joint | 8.60 | 25.6 | 25.4 |
| | Mono-Postr-Joint | **8.45** | **25.7** | 25.6 |
| | Prox-Postr-Joint | 8.52 | 25.6 | 25.5 |
| IWSLT15 EN-VI | Soft | 10.27 | 26.6 | 26.4 |
| | Sparse | 10.13 | 26.6 | 26.1 |
| | Prior-Joint | 9.67 | 27.4 | 27.3 |
| | Postr-Joint | **9.11** | **27.6** | 27.4 |
| | Mono-Postr-Joint | 9.52 | **27.6** | 27.3 |
| | Prox-Postr-Joint | 9.59 | 27.5 | 27.3 |
| IWSLT14 VI-EN | Soft | 8.30 | 24.7 | 24.6 |
| | Sparse | 8.48 | 24.2 | 23.9 |
| | Prior-Joint | 7.57 | 25.7 | 25.6 |
| | Postr-Joint | 7.34 | **25.9** | 25.8 |
| | Mono-Postr-Joint | **7.14** | **25.9** | 25.6 |
| | Prox-Postr-Joint | 7.26 | **25.9** | 25.9 |
| WAT17 JA-EN | Soft | 12.46 | 18.9 | 18.5 |
| | Sparse | 14.18 | 17.5 | 16.8 |
| | Prior-Joint | 10.00 | 20.6 | 20.2 |
| | Postr-Joint | 9.96 | 20.5 | 20.3 |
| | Mono-Postr-Joint | 9.98 | 20.7 | 20.5 |
| | Prox-Postr-Joint | **9.78** | **20.9** | 20.5 |

Table 1: Translation Perplexity and BLEU

Our results are in Table 1 where we show perplexity (PPL) and BLEU with beam size 4 and 10. All Postr-Joint variants and Prior-Joint outperform soft attention and sparse-attention by large margins.

Moreover models with posterior attention show improvement over those which use prior attention. We also observe small improvements over the more sophisticated and compute-intensive variational attention model likely due to the use of the exact posterior during inference instead. [1] This clearly shows the performance advantage of joint modeling and posterior attention. We shall analyze the reasons for these improvements later.

Next we explore the impact of different coupling models discussed in 2.3.1. For that focus on methods Postr-Joint, Prox-Postr-Joint, and Mono-Postr-Joint in Table 1. We obtain some gains over Postr-Joint by explicitly modeling attention coupling. For language-pairs with a natural monotonic alignment like German-English, Mono-Postr-Joint slightly outperforms other models by (0.1-0.2 BLEU points). English-Vietnamese is a more non-monotonic pair and as expected we do not find gains by incorporating a monotonic bias.

## 4.2 Morphological Inflection

To demonstrate the use of our model beyond translation, we next consider the task of generating morphological inflections. We use inflection forms for German Nouns (de-N) and German Verbs (de-V) from Durrett & DeNero (2013)'s. A model is trained separately for each type of inflection to predict the inflected character sequence. We train a one layer encoder and decoder with 128 hidden LSTM units each with a dropout rate of 0.2 using Adam and measure 0/1 accuracy. We also ran the 100 units wide two layer LSTM with hard-monotonic attention model Aharoni & Goldberg (2017) labeled Hard-Mono[2].

| Data | Soft | Hard-Mono | Prior-Joint | Postr-Joint | Mono-Postr-Joint | Prox-Postr-Joint |
|------|------|-----------|-------------|-------------|------------------|------------------|
| de-N | 85.50 | 85.65 | 85.81 | 85.88 | **86.87** | 85.81 |
| de-V | 94.91 | 95.31 | 95.52 | 95.5 | **95.71** | 95.4 |

Table 2: Test accuracy for morphological inflection

Using joint modeling we get significant gains (0.3 points) even against task-specific hard-monotonic attention, showing that our approach is more general than translation. Moreover when we use Mono-Postr-Joint which has a structural bias towards task-specific monotonic attention, we obtain immense improvements (upto 1 accuracy point) over joint models.

## 4.3 Explaining why we score above Soft Attention

We attempt to get more insights on why posterior attention models score over soft attention in end to end accuracy. We show that the main reason is better alignment of input and output because of a more precise attention model. We demonstrate that by first showing some anecdotes of better alignment, then showing that posterior attention is more focused (has lower entropy), provides better alignment accuracy, and better input coverage. For these runs we perform experiments in the teacher forcing setup so as to compare two models' distributions under identical inputs.

**Anecdotal Examples** Fig2 presents the heatmap of difference between Postr-Joint and Soft-Attention on two different sentences. In each figure the red regions represents where Postr-Joint has greater attention and blue where soft-attention has greater focus. One can observe that Soft-Attention is far more diffused. More importantly, we can see that Postr-Joint is able to correct mistakes and provides the appropriate context for the next step. For example in Fig2a Soft-Attention (blue) has maximum focus on the source word 'generationen' when the target word is innovation which corresponds to 'innovationen'; on the other hand Postr-Joint is able to correct this. Similarly while producing the phrase 'but the same' Postr-Joint focuses the attention on the source word 'dasselbe' Fig2b. This provides insight into as to how by providing better contexts via incorporating the target, posterior attention can outperform prior attention.

**Attention Entropy Vs Accuracy** We expect Soft-Attn to be worse hit by high attention uncertainty than other models. This, if true, could illustrate that $P(y_t|\mathbf{x}_t)$ distribution can be learned more easily

---

[1]Our numbers are different from the one reported by Deng et al. (2018) primarily because they used BPE encoding while we used word encoding. Using BPE too our model outperforms the variational model (**33.9** vs 33.7)

[2]https://github.com/roeeaharoni/morphological-reinflection

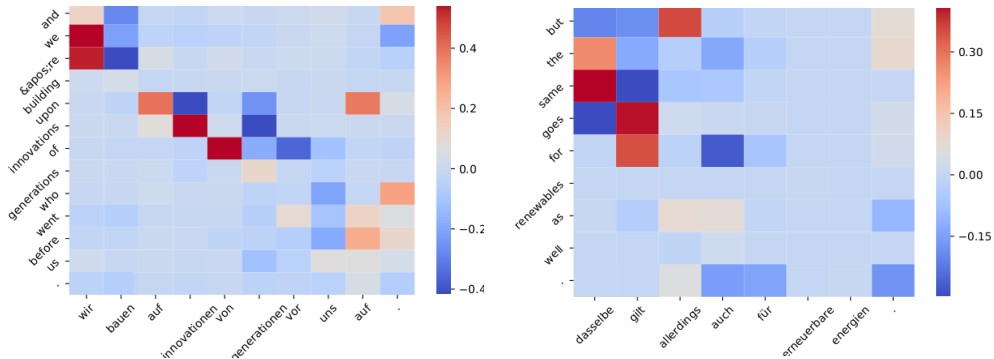

Figure 2: Heatmap of differences between Posterior-Attention (Red) and Soft-Attention (Blue). Mark the corrected red alignments for 'innovation' and 'but the same'

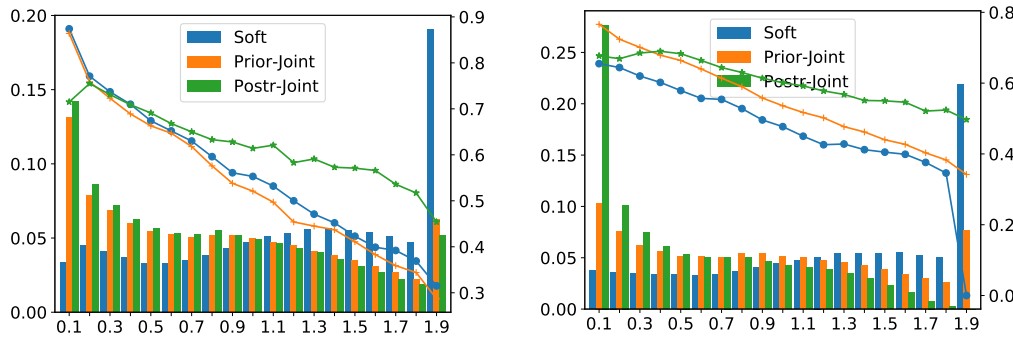

Figure 3: Variation of accuracy and histogram of attention entropy on De-En (left) and En-De (right) . Note the smoother accuracy decay in Postr-Joint and the entropy distibution for Sot-Attention

if the input is 'pure', rather than diffused via pre-aggregation. To this end we plot the accuracy of Postr-Joint, Prior-Joint and Soft-Attn under increasing attention entropy in Figure 3 on the English-German pair. As one can expect the accuracy drops off quickly as attention uncertainty rises. The plot also presents the histogram of the fraction of cases with different attention uncertainties. Soft attention models (blue) have significantly higher number of cases of high attention uncertainty, leading to low performance. One of the primary means by which joint models outperformed soft-attention is by reducing the number of such cases. These figures also provide insight into another mechanism by which posterior attention boosts performance. One can see that the accuracy drops off much more smoothly wrt attention uncertainty in posterior attention models (green). In fact in cases of high attention certainty (low attention entropy) Postr-Joint slightly underperforms Prior-Joint, however due to relatively stabler behavior gives better performance overall.

**Alignment accuracy** Failure of attention to produce latent structures which correspond to linguistic structures has been noted by Koehn & Knowles (2017); Ghader & Monz (2017).Based on few examples, we hypothesize that Posterior Attention should be able to produce better alignments. To test this we used the **RWTH German-English** dataset which provides alignment information manually tagged by experts, and compare the alignment accuracy for Soft, Prior-Joint and Postr-Joint attentions. Following the procedure in Ghader & Monz (2017) the most attended source word for each target word is taken as the aligned word. We used the AER metric Koehn (2010) to compare these against the expert alignments.

Table 3 presents the AER accuracy for different models. One can read off that Postr-Joint model beats the second best model ( Prior-Joint ) by more than 10%, and dwarfs soft-attention by a huge margin, proving that posterior alignments are significantly more compatible with true alignments.

| Attention | AER |
|---|---|
| Soft | 0.449 |
| Prior-Joint | 0.502 |
| Postr-Joint | **0.583** |

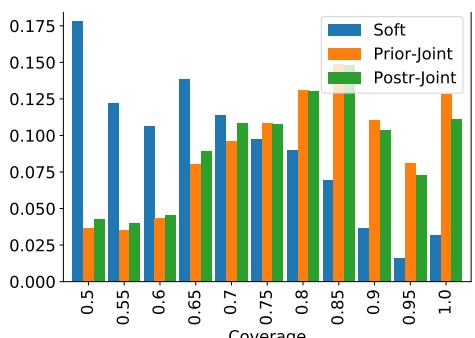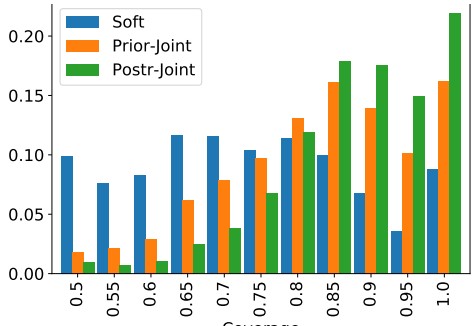

Figure 4: Coverage for different attention models on the En-De (left) and De-En(right) tasks

**Fraction of covered tokens**   A natural expectation for translation is that by the time the entire output sentence has been produced, attention would have covered the entire input sequence. A loss based on this precise heuristic was used in Chorowski & Jaitly (2016) to improve the performance of a attention based seq2seq model for speech transcription. In this experiment we try to indirectly assess reliability of different attention models via measuring whether cumulatively attention has focused on the entire input sequence.

We plot the frequency distribution of the coverage in Fig4. Note that in soft attention model, there are many sentences which do not receive enough attention during the entire decoding process. Prior-Joint and Postr-Joint have similar behavior with few instances of one outperforming the other, however both outperform soft attention by huge margins.

## 5   CONCLUSION

We show in this paper that none of the existing attention models adequately model the dependence of the output and attention along the length of the output for general sequence prediction tasks. We propose a factorization of the joint distribution, and develop practical approximations that allows the joint distribution to decompose over output tokens, much like in existing attention. Our more principled probabilistic joint modeling of the dependency structure leads to three important differences. First, the output token distribution is obtained by aggregating predictions across all attention. Second, the concept of conditioning attention on the current output i.e. a *posterior* attention for inferring the next output becomes important. Our experiments show that it is sounder, more meaningful and more accurate to condition subsequent attention distribution on the posterior attention. Thirdly, via directly exposing attention coupling, we have a principled way to directly incorporate task-specific structural biases and prior knowledge into attention. We experimented with some simple biases and found boosts in related tasks. Our work opens avenues for future work in scaling these techniques to large-scale models and multi-headed attention. Another promising line is to incorporate more complex biases like phrasal structure or image segments into joint attention models.

**Acknowledgements**   We thank NVIDIA Corporation and Flipkart for supporting this research.

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

## APPENDIX

Figure 5: Individual attention distribution for some sentences, Postr-Jointon left and Soft-Attn on the right

