# OpenReview forum: "Posterior Attention Models for Sequence to Sequence Learning"
_ICLR.cc/2019/Conference_

### Official Review · AnonReviewer2 · 2018-10-26
**This paper presents a novel posterior attention model for seq2seq problems. The PAM exploits the dependencies among attention and output variables, unlike existing attention models that only gives ad-hoc design of attention vectors. The experiments demonstrate their claimed advantages.**

**Rating:** 7
**Confidence:** 4

**Review:**

Pros:
1. This work presents a novel construction of the popularly-used attention modules. It points out the problems lied in existing design that attention vectors are only computed based on parametric functions, instead of considering the interactions among each attention step and output variables. To achieve that, the authors re-write the joint distribution as a product of tractable terms at each timestamp and fully exploit the dependencies among attention and output variables across the sequence. The motivation is clear, and the proposed strategy is original and to the point. This makes the work relative solid and interesting for a publication. Furthermore, the authors propose 3 different formulation for prior attention, making the work even stronger.
2. The technical content looks good, with each formula written clearly and with sufficient deductive steps. Figure 1 provides clear illustration on the comparison with traditional attentions and shows the advantage of the proposed model.
3. Extensive experiments are conducted including 5 machine translation tasks as well as another morphological inflection task. These results make the statement more convincing. The authors also conducted further experiments to analyze the effectiveness, including attention entropy evaluation.

Cons:
1. The rich information contained in the paper is not very well-organized. It takes some time to digest, due to some unclear or missing statements. Specifically, the computation for prior attention should be ordered in a subsection with a section name. The 3 different formulations should be first summarized and started with the same core formula as (4). In this way, it will become more clear of where does eq(6) come from or used for. Currently, this part is confusing.
2. Many substitutions of variables take place without detailed explanation, e.g., y_{<t} with s_t, a with x_{a} in (11) etc. Could you explain before making these substitutions?
3. As mentioned, the PAM actually computes hard attentions. It should be better to make the statement more clear by explicitly explaining eq(11) on how it assembles hard attention computation.

QA:
1. In the equation above (3) that computes prior(a_t), can you explain how P(a_{t-1}|y_{<t}) approximates P(a_{<t}|y_{<t})? What's the assumption?
2. How is eq(5) computed using first order Taylor expansion? How to make Postr inside the probability? And where does x_a' come from?
3. Transferring from P(y) on top of page 3 to eq(11), how do you substitute y_{<t}, a_t with s_t, x_j? Is there a typo for x_j?
4. Can you explain how is the baseline Prior-Joint constructed? Specifically, how to compute prior using soft attention without postr?

---

> ### Author Response · Authors · 2018-11-27
> **Author response to reviewer**
>
> We thank the reviewer for their feedback.
> We have rewritten the derivation of our factorization and made the assumptions clearer in Section 2.2 .
> Section 2.2.1 has also been revised describing the different variants and their intuition, deriving them all from Eqn 4.
> We have also fixed some notational discrepancies as pointed out by the reviewer for which we are thankful.
>
> QA
> 1)
> We have rewritten that section, but the simplification comes about because of the Markovian assumption that P(a_t|a_{<t}) = P(a_t|a_{t-1}).  This makes \sum_{a_{t-1}} P(a_t|a_{<t})P(a_{<t}|y_{<t})  =  \sum_{a_{t-1}} P(a_t|a_{<t}) P(a_{t-1}|y_{<t}).
>
>
> 2)
> The Taylor trick was used by [1] to simplify the expectation computation. Essentially if the average value of a function is computed at different points, one can compute the Taylor expansion of the function at average of the points leaving only second order terms.
>
> \Sigma f(x_i) = \Sigma f( xm + x_i - xm) = \Sigma [ f(xm) + f’(xm)(x_i - xm) + second order terms ] = \Sigma f(xm) + df(xm)\Sigma(x_i - xm) + second order =  \Sigma f(xm) + df(xm)*0 + second order \approx \Sigma f(xm)
>
> 3)
> s_t is the decoder state after feeding in output y_{t-1} and attention at step {t-1}. Like in standard seq2seq literature, we rely on the decoding RNN state to capture the dependence on history of output tokens. Under the assumption that y_t depends directly on attention 'a' at t and previous tokens, we use the decoder state s_t and the encoder state x_{a}. Indeed as pointed 'j' was a typo.
>
> 4)
> The main difference between the prior-joint and postr-joint model is which attention gets propagated further down. The prior-joint model behaves analogously to the standard soft-attention in ignoring any interaction between output and attention. In fact, it is a version of an IBM model 1. We have expanded on this in Section3 paragraph 7 and Section4 paragraph 1
>
> [1] Xu et al; Show, attend and tell: Neural image caption generation with visual attention , 2015

---

### Official Review · AnonReviewer3 · 2018-11-01
**Posterior attention improves sequence to sequence learning**

**Rating:** 9
**Confidence:** 4

**Review:**

Originality: Existing attention models do not statistically express interactions among multiple attentions. The authors of this manuscript reformulate p(y|x) and define prior attention distribution (a_t depends on previous outputs y_<t) and posterior attention distribution (a_t depends on current output y_t as well), and essentially compute the prior attention at current position using posterior attention at the previous position. The hypothesis and derivations make statistical sense, and a couple of assumptions/approximations seem to be mild.

Quality: The overall quality of this paper is technically sound. It pushs forward the development of attention models in sequence to sequence mapping.

Clarity: The ideas are presented well, if the readers go through it slowly or twice. However, the authors need to clarify the following issues:
x_a is not well defined.
In Section 2.2, P(y) as a short form of Pr(y|x_1:m) could be problematic and confusing in interpretation of dependency over which variables.
Page 3: line 19 of Section 2.2.1, should s_{n-1} be s_{t-1}?
In Postr-Joint, Eq. (5) and others, I believe a'_{t-1} is better than a', because the former indicate it is attention for position t-1.

I am a bit lost in the description of coupling energies. The two formulas for proximity biased coupling and monotonicity biased coupling are not well explained.

In addition to the above major issues, I also identified a few minors:
significant find -> significant finding
Last line of page 2: should P(y_t|y_<t, a_<n, a_n) be P(y_t|y_<t, a_<t, a_t)?
top-k -> top-K
a equally weighted combination -> an equally weighted combination
Some citations are not used properly, such as last 3rd line of page 4, and brackets are forgotten in some places, etc.
End of Section 3, x should be in boldface.
non-differentiability , -> non-differentiability,
Full stop "." is missing in some places.
Luong attention is not defined.

Significance: comparisons with an existing soft-attention model and an sparse-attention model on five machine translation datasets show that the performance of using posterior attention indeed are better than benchmark models.

Update: I have read the authors' response. My current rating is final.

---

> ### Author Response · Authors · 2018-11-27
> **Author response to reviewer**
>
> We thank the reviewer for the comments.
> In light of comments about some of the notation and description from all reviewers, we have revised the model description considerably. We have also fixed some notational inconsistencies as pointed out.
>
> We have also revised Section 2.2.1 to better explain the formula and intuition of the coupling energies.

---

### Official Review · AnonReviewer1 · 2018-11-03
**Very interesting contribution**

**Rating:** 8
**Confidence:** 5

**Review:**

This paper proposes a new sequence to sequence model where attention is treated as a latent variable, and derive novel inference procedures for this model. The approach obtains significant improvements in machine translation and morphological inflection generation tasks. An approximation is also used to make hard attention more efficient by reducing the number of softmaxes that have to be computed.

Strengths:
- Novel, principled sequence to sequence model.
- Strong experimental results in machine translation and morphological inflection.
Weaknesses:
- Connections can be made with previous closely related architectures.
- Further ablation experiments could be included.

The derivation of the model would be more clear if it is first derived without attention feeding: The assumption that output is dependent only on the current attention variable is then valid. The Markov assumption on the attention variable should also be stated as an assumption, rather than an approximation: Given that assumption, as far as I can tell the (posterior) inference procedure that is derived is exact: It is indeed equivalent to the using the forward computation of the classic forward-backward algorithm for HMMs to do inference.
The model’s overall distribution can then be defined in a somewhat different way than the authors’ presentation, which I think makes more clear what the model is doing:
p(y | x) = \sum_a \prod_{t=1}^n p(y_t | y_{<t}, x, a_t) p(a_t | y_{<t}, x_ a_{t-1}).
The equations derived in the paper for computing the prior and posterior attention is then just a dynamic program for computing this distribution, and is equivalent to using the forward algorithm, which in this context is:
 \alpha_t(a) = p(a_t = a, y_{<=t}) = p(y_t | s_t, a_t =a) \sum_{a’} \alpha_{t-1}(a’) p(a_t = a | s_t, a_{t-1} = a’)

The only substantial difference in the inference procedure is then that the posterior attention probability is fed into the decoder RNN, which means that the independence assumptions are not strictly valid any more, even though the structural assumptions are still encoded through the way inference is done.
[1] recently proposed a model with a similar factorization, although that model did not feed the attention distribution, and performed EM-like inference with the forward-backward algorithm, while this model is effectively computing forward probabilities and performing inference through automatic differentiation.

The Prior-Joint variant, though its definition is not as clear as it should be, seems to be assuming that the attention distribution at each time step is independent of the previous attention (similar to the way standard soft attention is computed) - the equations then reduce to a (neural) version of IBM alignment model 1, similar to another recently proposed model [2]. These papers can be seen as concurrent work, and this paper provides important insights, but it would strengthen rather than weaken the paper to make these connections clear.

The results clearly show the advantages of the proposed approach over soft and sparse attention baselines. However, the difference in BLEU score between the variants of the prior or posterior attention models is very small across all translation datasets, so to make claims about which of the variants are better, at a minimum statistical significance testing should be done. Given that the “Prior-Joint” model performs competitively, is it computationally more efficient that the full model?

The main missing experiment is not doing attention feeding at all. The other experiment that is not included (as I understood it) is to compute prior and posterior attention, but feed the prior attention rather than the posterior attention.

The paper is mostly written very clearly, there are just a few typos and grammatical errors in sections 4.2 and 4.3.

Overall, I really like this paper and would like to see it accepted, although I hope that a revised version would make the assumptions the model is making clearer and make connections to related models clearer.

[1] Neural Hidden Markov Model for Machine Translation, Wang et al, ACL 2018.
[2] Hard Non-Monotonic Attention for Character-Level Transduction, Wu, Shapiro and Cotterell, EMNLP 2018.

---

> ### Author Response · Authors · 2018-11-27
> **Author response to reviewer**
>
> We thank the reviewer for the feedback.  We have discussed the papers mentioned by you and other reviewers in the Related work section, and also added new empirical comparisons.
>
> We are also very grateful for suggesting the alternative derivation.  We have added a discussion regarding your suggestion in Section 2.4 .  We have also simplified our derivation by explicitly stating and pulling up the
> Markov assumption about attention dependencies earlier.
>
>
> The prior joint model is indeed related to a neural IBM model 1, and has been used in multiple recent works as also pointed out by Yoon Kim.
>
> From an efficiency perspective, the various posterior attention models are only marginally slower than prior-joint which does the more compute-intensive part of calculating P(y_t) for each of the top-K attentions.  Thereafter, for tasks like translation, the coupled attention computation almost comes for "free". In fact, we observed no measurable difference in the average time per step between the two models
>
> Most seq2seq models rely upon attention feeding at all timesteps, and so we had not experimented with that model. We are providing some of the results of the experiment in the response here.
>  Dataset      B=4  B=10
> de-en          28.8  28.6
> en-de          24.0  23.9
> en-vi            26.9  26.6
>
> These numbers are roughly on par with soft-attention and show the importance of feeding the attention context.
>
> We also ran some experiments with the suggestion of feeding the prior attention, which are as follows
>             B=4     B=10
> en-vi    27.3    27.0
> vi-en    25.7    25.7
>
> These results are similar to or slightly worse than the prior-joint model. We are currently in the process of evaluating this on more tasks.

---

> > ### Comment · AnonReviewer1 · 2018-12-03
> > **Thanks for your response**
> >
> > I thank the authors for improving the clarity of the model derivation and updating the paper to mention related work and alternative derivations. I agree that the author's formulation provides novel and interesting insights. However, I would just like the final version of the paper to be more explicit - preferable in both the introduction and model derivation - about the relation of their models to the latent/hard attention models that have been discussed here. Just mentioning these papers in the related work section is not sufficient to fully contextualize this work (as was asked for by the other reviewers and commenters as well). Mentioning that these models are essentially neural generalizations of the classical IBM alignment models (Brown et al., 1993) is also helpful for contextualization.

---

> > > ### Author Response · Authors · 2018-12-04
> > >
> > > Thanks for the suggestion. We will take this into account and contextualize better in the next draft.

---

### Public Comment · ~Yoon_Kim1 · 2018-11-06
**one question/comment**

Hi there, thanks for a very nice paper. It is great to see that posterior inference substantially increases alignment accuracy! I also liked the application of the model across a diverse range of languages/tasks.

I had one quick question, and one comment:

Question:
- How do you differentiate through the top-K approximation? Do you use the straight through estimator? How much faster was top K vs actually enumerating?

Comment:
- There are several recent works that have also formalized attention as a latent variable and have exactly/approximately optimized the log marginal likelihood. It would be great to see this work put in context of existing work!

Wu et al. Hard Non-Monotnic Attention for Character-Level Transduction. EMNLP 2018
Shankar et al. Surprisingly Easy Hard-Attention for Sequence to Sequence Learning. EMNLP 2018.
Deng et al. Latent Alignment and Variational Attention. NIPS 2018.

---

> ### Public Comment · (anonymous) · 2018-11-13
> **Empirical Comparison to (Deng et al., 2018) ?**
>
> Yes it'd be nice to see a comparison of this work to (Deng et al., 2018) which also models attention as a latent variable and has released code here: https://github.com/harvardnlp/var-attn

---

> > ### Author Response · Authors · 2018-11-27
> >
> >
> > 1)
> > Yes, we have used the straight through estimator. On our larger datasets we were not able to do full enumeration because of memory constraint.  For En-Vi we can run the exact enumeration and for that task the top-k marginalization reduced time per-step by around 50\%  (0.354s vs 0.655s per step) and the required memory by a factor of 4 with very minor impact on BLEU
> >
> > 2)
> > We thank you for giving pointers to related work. The reviewers also pointed similar works. We have discussed them in the Related Work section of the revised version.  Also, we have included some experimental comparisons with all of these.

---

> > > ### Public Comment · ~Yoon_Kim1 · 2018-11-27
> > > **thanks!**
> > >
> > > Thanks for the detailed response!

---

### Public Comment · (anonymous) · 2018-11-14
**More high level insights needed**

The contribution is interesting, but besides the experimental part is a little bit too dry. The paper would immensely benefit of a more high level description and insights about the architecture proposed, as well as a graphical representation (such as a block diagram) to make the architecture understandable at a first glance.

---

> ### Author Response · Authors · 2018-11-27
>
> We have rewritten Section 2.2 of the paper, which simplifies the presentation and makes the need for posterior attention more obvious. The network architecture and connections are the same as standard soft attention model. The difference is entirely on how attention is computed.

---

### Meta-Review · Area_Chair1 · 2018-12-13
**One of the better papers at the conference**

**Confidence:** 5
**Recommendation:** Accept (Poster)

**Metareview:**

The reviewers of this paper agreed that it has done a stellar job of presenting a novel and principled approach to attention as a latent variable, providing a new and sound set of inference techniques to this end. This builds on top of a discussion of the limitations of existing deterministic approaches to attention, and frames the contribution well in relation to other recurrent and stochastic approaches to attention. While there are a few issues with clarity surrounding some aspects of the proposed method, which the authors are encouraged to fine-tune in their final version, paying careful attention to the review comments, this paper is more or less ready for publication with a few tweaks. It makes a clear, significant, and well-evaluate contribution to the field of attention models in sequence to sequence architectures, and will be of great interest to many attendees at ICLR.